# Advancements in Transcatheter Aortic Valve Implantation: A Focused Update

**DOI:** 10.3390/medicina57070711

**Published:** 2021-07-14

**Authors:** Niccolò Ciardetti, Francesca Ciatti, Giulia Nardi, Francesca Maria Di Muro, Pierluigi Demola, Edoardo Sottili, Miroslava Stolcova, Francesca Ristalli, Alessio Mattesini, Francesco Meucci, Carlo Di Mario

**Affiliations:** Structural Interventional Cardiology, Department of Clinical and Experimental Medicine, Clinica Medica, Room 124, Careggi University Hospital, Largo Brambilla 3, 50139 Florence, Italy; niccolo.ciardetti@unifi.it (N.C.); ciattifrancesca@gmail.com (F.C.); giulia.nardi.fi@gmail.com (G.N.); fdimuro94@gmail.com (F.M.D.M.); pierluigidemola@gmail.com (P.D.); edo.sottili@gmail.com (E.S.); mira.stolcova@hotmail.com (M.S.); f.ristalli@hotmail.it (F.R.); amattesini@gmail.com (A.M.); francescomeu19@gmail.com (F.M.)

**Keywords:** transcatheter aortic valve implantation, TAVI, aortic valve replacement, iliac intravascular lithotripsy, cerebral protection devices, optimal valve positioning, valve-in-valve

## Abstract

Transcatheter aortic valve implantation (TAVI) has become the leading technique for aortic valve replacement in symptomatic patients with severe aortic stenosis with conventional surgical aortic valve replacement (SAVR) now limited to patients younger than 65–75 years due to a combination of unsuitable anatomies (calcified raphae in bicuspid valves, coexistent aneurysm of the ascending aorta) and concerns on the absence of long-term data on TAVI durability. This incredible rise is linked to technological evolutions combined with increased operator experience, which led to procedural refinements and, accordingly, to better outcomes. The article describes the main and newest technical improvements, allowing an extension of the indications (valve-in-valve procedures, intravascular lithotripsy for severely calcified iliac vessels), and a reduction of complications (stroke, pacemaker implantation, aortic regurgitation).

## 1. Introduction. Extended Indications: A Powerful Stimulus to Reduce All Complications

Aortic valve replacement (AVR) represents the only effective treatment to reduce mortality in symptomatic patients with severe aortic stenosis [1]. Surgical AVR (SAVR) was the gold standard in most patients until a few years ago, with transcatheter aortic valve implantation (TAVI) being relegated only in the setting of patients with increased surgical risk. It is now clear that TAVI is non-inferior to SAVR also in low-risk patients, allowing lower mortality rates, strokes, major bleedings, atrial fibrillation, and shorter hospital length of stay and recovery time [1,2]. These advantages increased the attractiveness of TAVI to the extent that, in the last five years in the USA, the number of TAVI procedures overcame both isolated SAVR and all SAVR [3]. This radical shift in clinical practice is attributable to a virtuous cycle (Figure 1) generated by increased operators’ experience on one side and improved technology on the other. The ever-growing number of TAVI procedures benefit from improved pre-procedural assessment, with dedicated protocols of computed tomography (CT) image acquisition and reconstruction, including sophisticated but still experimental 3D software simulating procedural outcome as severity of aortic regurgitation and need of pacemaker (PM) implantation (FEops HEART Guide™, Gent, Belgium) [4]. The procedural technique itself has been refined and aims, in selected patients, to a ‘minimalistic’ approach, which consists of percutaneous femoral primary access route–now used in more than 95% of patients [3] secondary radial access for supravalvular aortic angiography to guide valve deployment and check optimal closure of the primary access [5], direct implantation without pre-dilatation [6], vascular and transthoracic echocardiography with no transoesophageal echocardiography and the increasing use of conscious sedation and local anaesthesia [7]. Engineering advances on transcatheter valves’ design led to catheter miniaturization and lower paravalvular leaks (PVL), thanks to optimized sealing “skirts”, applied on both balloon-expandable and self-expanding valves. Major vascular complications were significantly reduced [3] by improvements in prosthesis delivery systems, which typically come with thinner profiles of 14–16 French, and by the use of advanced vascular closure devices (e.g., two Perclose Proglide™ [Abbott Medical, Santa Clara, CA, USA] w/wo additional Angio-Seal™ [Terumo Corporation, Tokyo, Japan], Manta^®^ Vascular Closure Device [Teleflex, Wayne, PA, USA]). The application of intravascular lithotripsy in iliac arteries (Shockwave Medical Inc, Santa Clara, CA, USA) offers the possibility to perform transfemoral TAVI also in patients with severe peripheral calcifications, avoiding other less-favourable routes [8]. Cerebral embolic prevention devices are tools that can be used to reduce stroke, one of the most feared complications of AVR now dramatically reduced in recent trials. Higher implantation technique, the adoption of optimal projections (e.g., ‘cusp-overlap’ approach) and identification of patients with increased risk (e.g., pre-existing right bundle branch block) are only some of the features that we need to consider to decrease permanent PM implantation rates [9,10].

Improvements of clinical endpoints in TAVI-treated patients are hard evidence of the above-described virtuous cycle, as shown by a mortality rate at 30 days, life-threatening/disabling bleedings and strokes, reduced from 7.2% to 2.5%, 6.3% to 1.8% and 2.1% to 1.6%, respectively [3].

The evolution of TAVI needs a readjustment of AVR indications by the main international scientific societies, with the last ESC/EACTS guidelines [11] on valvular heart disease being now outdated and due to be reissued in August 2021. The recently published ACC/AHA guidelines [12] contain a drastic change of direction compared to the previous edition. The Heart Team evaluation and estimated surgical risk remain a key element to decide between SAVR or TAVI, with the patient’s age and life expectancy as other main variables to be considered and provided that the decision is shared with the patient and the TAVI is doable transfemorally.

Nevertheless, the Heart Team’s role is not over: indeed, it will continue to be fundamental to identify and weigh comorbidities often associated with severe aortic stenosis, to evaluate life expectancy irrespective of chronological age, to better address the treatment choice in patients excluded from the main randomised clinical trials and, not least, to avoid futility especially in elderly patients with cognitive impairment or poor quality of life irrespective of TAVI [13].

Since there are still limited data on transcatheter valve durability beyond five years, TAVI in younger patients is not indicated yet. Anyway, valve-in-valve procedures will have a key role, as shown by the effort in creating surgical valves specifically designed to improve valve-in-valve procedures and to reduce patient–prosthesis mismatch (e.g., Edwards Inspiris Resilia [Edwards Lifesciences, Irvine, CA, USA]).

If transcatheter valve durability will be confirmed to be the same as biological surgical valves, TAVI will supersede almost completely SAVR, even if surgery will remain the gold standard in selected cases (mechanical valves, endocarditis, combined surgery on other valves, aortic aneurysms, severe coronary artery disease and unfavourable valve/access anatomy).

The article will review and discuss the main and most recent technical advancements of transfemoral TAVI even in more complex patients, aiming to reduce peri-procedural strokes, the rate of pacemaker implantation and post-TAVI aortic regurgitation; the novelties regarding valve-in-valve and valve-in-TAVI procedures will be reviewed as well.

## 2. Novelties in Femoral Approach: From US Guided Micro-Puncture to Iliac Intravascular Lithotripsy

Since the first percutaneous heart valve was implanted in 2002 by Cribier et al. [14], the technology has been improved over the years to be minimally invasive and maximally effective [15]. Numerous factors such as a smaller delivery system, the introduction of novel percutaneous vascular closure devices, a more accurate patient and access selection and increased operator experience have contributed to reducing vascular complications [16]. Toggweiler et al. in their initial practice of transfemoral TAVI in high-risk patients reported that the major source of morbidity was related to vascular access and underlined the importance of accurate pre-procedural screening and improved vascular management in reducing vascular morbidity and mortality risk [17]. The STS-ACC registry annual report recently published the 30-day major vascular access site complications for high-risk patients had declined to 1.8% in 2019 with a predominant use of trans-femoral (TF) access [3].

Currently, the demonstration of efficacy and safety of TAVI even in low surgical risk patients contributed to further extend TAVI indications and the size of the population that could benefit from this percutaneous treatment.

However, as commonly reported in large, randomised trials and registries, the TF access has become the preferred initial/first option approach for TAVI delivery and the ability to treat the 95% of patients with transfemoral access has been associated with lower bleeding rates, reduced hospitalization, prompt ambulation, and discharges to home [18]. The transfemoral approach is also well tolerated under local anaesthesia and is compatible with the successful implantation of closure devices (suture-based Proglide™ and collagen-based Manta^®^ plugs).

According to previous literature, access site bleeding represents one of the most common major vascular complications and is frequently associated with an ominous prognosis, especially when requiring red blood cells (RBC) transfusions [19,20].

In the PRAGMATIC study, one of the largest series of patients treated with TAVI, patients requiring RBC transfusion had an increased risk of mortality at 1 year and an increased risk of major stroke and acute kidney injury, compared to patients who did not require RBC transfusions [21].

In the last decade, different tools have revealed their usefulness in preventing the development of peri-procedural vascular complications, such as ultrasound (US) guided micro-puncture and peripheral intravascular lithotripsy.

In the current practice, the standard of care of access cannulation is represented by blind puncture under palpation at most confirming height with fluoroscopic guidance. After achieving the secondary arterial access (mainly left radial) the operators proceed to identify with US and fluoroscopy the anatomical landmarks (femoral head and bifurcation) to cannulate the main TAVI access. Especially in the case of vascular calcification, an accurate selection of the cannulation site can offer great advantages in avoiding anterior calcification and successfully implanting percutaneous closure devices. This may reduce vascular complications, more frequently achieves successful haemostasis and improves clinical outcomes. US guidance, indeed, allows a real-time examination of the vessel wall and the selection of the ideal puncture zone by identifying conventional landmarks such as the femoral bifurcation (below) between the superficial femoral artery and the profunda femoris and the inguinal ligament (upper). The ideal cannulation site is included in the horizontal segment of the common femoral artery (CFA), in the middle of the free-calcium anterior wall. This technique demonstrated to improve puncture success rate at the first attempt and to reduce accidental venipuncture rate. Furthermore, the US-guided approach allows operators to accurately control the puncture site of the CFA, increasing physician confidence and reducing patient’s life-threatening or retroperitoneal complications [22]. The use of dedicated micro-puncture 21-gauge (G) needles with the tip more visible with the US reduces the rate of vascular complications with a significant decrease in the number of groin hematomas compared to standard large bore needles [23].

These results have also been confirmed by the FEMORIS trial that aimed to compare micro-puncture with 21 G versus standard 18 G needle in more than 400 patients [24].

Recently, an observational study demonstrated a reduction in the composite endpoint of RBC transfusion and vascular and bleeding complications using a US-guided approach in transfemoral TAVI [25].

In 2020 Vincent et al. used propensity score matching to confirm a strong clinical benefit of US guidance in percutaneous trans-femoral TAVI. In this study, they demonstrated that vascular and bleeding complications including life-threatening or major bleedings were reduced in the US-guided puncture group compared with the fluoroscopy-guided access cannulation arm [22].

Data derived from coronary percutaneous coronary intervention (PCI) scenario have supported this procedural strategy.

The sub-analysis of the SAFE-PCI (Study of Access Site for Enhancement of PCI for Women) demonstrated that the use of micro-puncture and US-guided puncture of the femoral artery was associated with similar bleeding events or vascular complications as radial access for PCI [26]. Similarly, the multicentre, randomised FAUST (Femoral Arterial Access with UltraSound Trial) trial compared US versus fluoroscopic-guided CFA cannulation in procedures requiring a small size sheath. The US-guided approach reduced the number of attempts and vascular complications in femoral arterial access [27].

In recent years, the US-guided puncture approach is underused and adopted only in a minority of cases in clinical practice [28]. Since the latest-generation TAVI delivery systems require large-bore vascular access, recent findings support the use of US in reducing not only major vascular complications but also mean fluoroscopic time and a concomitant significant decrease of the radiation dose. Thus, intraprocedural US is likely to become the future standard of care in TAVI procedure for CFA cannulation.

If the presence of femoral calcium generally represents an obstacle to identify the adequate puncture site and to deploy a percutaneous closure device system, severe iliac arterial calcification together with tortuosity can also preclude a safe TF access and force operators to shift to alternative routes. This is not a rare condition if considering that peripheral stenoses, tortuosity and vessel calcifications affect 35% of the elderly population undergoing TAVI procedures [29]. This high prevalence is not unexpected since aortic valve degeneration and peripheral occlusive disease share the same pathophysiological substrate [30]. A narrowed luminal diameter with circumferential calcification is of particular relevance because non-calcified arteries may be stretched and successful insertion can be achieved with a lumen as small as 75% of the TAVI sheath’s outer diameter. For calcified tortuous arteries it is recommended that the lumen is at least 1.25 mm bigger than the sheath. For the 14 or 16 Fr inner diameter sheaths of the contemporary miniaturised delivery systems, this equates to minimal diameters of 6 to 7 mm in non-calcified and calcified vessels respectively [31]. In this scenario, the introduction of peripheral intravascular lithotripsy offers great advantages in preserving TF access for TAVI procedures.

Shockwave lithotripsy technology was introduced in the 80s for the treatment of urinary stones [32,33] and took advantage of the electro-hydraulically generated sonic pressure waves to selectively fracture calcium without damaging surrounding soft tissue. The use of electrohydraulic technology on a semi-compliant angioplasty balloon is based on the presence of two or more electrodes on the distal end of the balloon shaft, which generate circumferential shockwaves that effectively modify both intimal and medial calcium sparing soft tissue. This encouraging success in increasing vessel compliance led to the application of this technique in the context of complex vascular calcifications at the access site before TAVI (Figure 2). The efficacy and safety of intravascular lithotripsy (IVL) have been widely demonstrated in both coronary and peripheral arteries [34,35,36,37]. The recently published DISRUPT PAD III [37] trial represents the first randomised controlled trial comparing IVL with standard treatment. IVL was found to be superior to conventional percutaneous transluminal angioplasty (PTA) for the treatment of heavily calcified femoral-popliteal arteries. IVL safely reduced the percentage diameter stenosis but also the frequency and severity of vascular dissection and the need for further post-dilatation and stent implantation. Thanks to these promising results, IVL’s use has progressively expanded to other fields, such as facilitating the insertion of large-dimension sheaths in the context of TAVI procedures or mechanical circulatory support devices (IMPELLA, V-A ECMO) [38].

Two disposable catheters type exist for peripheral vascular use: the Shockwave IVL catheter M5 and S4 (Shockwave Medical Inc., Santa Clara, CA, USA). They differ in terms of the number of lithotripters inserted inside the shaft of the semi-compliant balloon, that is five and four respectively for peripheral use. The intravascular lithotripsy catheter is connected with a generator, programmed to deliver a predefined number of impulses at a rate of one per second with a maximum number of pulses of 160 for the peripheral smaller Shockwave S4 (generally used to treat femoral-popliteal calcification) and 300 for the peripheral larger Shockwave M5, more frequently used in iliac and CFA arteries lesions. For Shockwave M5, the balloons available have a diameter between 3.5 and 7 mm, a crossing profile of 0.054–0.073″ a length of 60 mm, can be used with guiding catheters of 6–7 French and are compatible with common 0.014″ guidewires [33].

After insertion of 0.014″ guidewire, an appropriate IVL balloon is selected and delivered across the calcified segment in the iliac and common femoral arteries to facilitate insertion of the TAVI delivery system. After balloon inflation at low pressure (3–4 Atmospheres), multiple activations are generally performed. The shockwaves delivered at low pressure modify the vessel stiffness with the creation of multiple longitudinal and transversal calcium fractures as demonstrated with optical coherence tomography in coronary lithotripsy sub-studies [39].

Based on the pre-operative lower limb CT angiography, IVL can be used for lesion preparation as an elective or bail-out strategy in patients with severe peripheral artery disease intended for TAVI but considered ineligible for standard transfemoral access.

After the first description of a case of IVL-assisted TAVI performed in December 2017 in Florence [40], only one prospective registry of fewer than 50 patients has been reported [8].

No studies have prospectively addressed the advantage conferred by the systematic use of this technique in reducing the need for non-TF approaches and further studies are needed to standardise this technology in routine clinical practice.

## 3. Cerebral Protection Devices for Stroke Prevention

Cerebrovascular events (CVEs) after TAVI are a rare but feared complication that may present as stroke, transient ischemic attack (TIA) or silent neurological event, i.e., absence of clinical symptoms despite acute ischemic cerebral lesion shown with neuroimaging or pathology.

CVEs are associated with increased 1-year mortality [41,42], are independent predictors of morbidity during the follow-up and have a major impact on quality of life, by impairing cognitive function and daily abilities [43,44]. CVEs are largely underdiagnosed, mainly due to the absence of a standardized definition and classification [44], with a thorough neurological examination not part of the routine check-ups after TAVI.

Compared to SAVR, stroke risk is significantly lower in TAVI procedures, especially in low-risk patients. In TAVI cohorts, stroke incidence at 30 days was 0.6% in PARTNER 3 and 0.5% in the Evolut-Low Risk Low Risk, while SAVR was associated with 2.4% and 1.7% rate of stroke in the two trials respectively [45,46]. Likely, the reduced risk of new-onset atrial fibrillation in patients undergoing TAVI affects the stroke risk, but several other factors are involved in the development of ischemic brain damage during surgery (aortic clamp, low-pressure perfusion during extracorporeal circulation, etc). Although the incidence of atrial fibrillation (AF) after TAVI is lower than after SAVR [45,46] and the pathophysiological link between the development of AF and TAVI is questionable, the new onset of AF, especially in frail and comorbid patients, represents an important issue to be considered for the choice of the antithrombotic therapy. In this regard, a recent open-label multicentre randomised trial [47] showed that anticoagulant therapy alone—either with vitamin K antagonist or direct-acting oral anticoagulant–reduces the risk of serious bleeding compared to anticoagulation plus clopidogrel with a similar rate of ischemic stroke and without an increase in thromboembolic complications.

The risk of CVE after TAVI may appear very low but, as indications expand to younger and lower-risk patients, the prevention of stroke becomes even more important [12]. Various cerebral protection devices (CPD) have been developed to reduce the risk of embolization of debris and/or thrombus during the procedure (Table 1). Since their use increases procedure time and costs, it is essential to identify which patients are at greater risk of CVEs to tailor the procedure to the patient. Indeed, risk factors could be patient-specific (chronic kidney disease, mitral stenosis, pre-existing atrial fibrillation, carotid artery disease), linked to valvular anatomical characteristics (patients with bicuspid valves have higher stroke rates in the first 30 days [48]), or related to the procedure itself.

The majority of acute strokes are ischemic (95%) and mainly secondary to peri- and intraprocedural factors. The use of stiff guidewires, bioprosthesis post-dilatation, large calliper TAVI delivery systems and prolonged procedural times have been reported to increase the intra-procedural stroke risk [49]. Haemodynamic instability, due to rapid ventricular pacing or anaesthetic drugs, could contribute to cerebral hypoperfusion and distress during the procedure. Access choice is an important feature that must be considered, as a transthoracic (transapical or transaortic) approach or even more trans-subclavian/axillary accesses have higher rates of disabling strokes compared to transfemoral access [50,51]. Valve type may also influence the stroke risk, and the SOLVE-TAVI randomised trial observed numerically higher stroke rates in the SAPIEN 3 group compared to the Evolut R arm (4.7% vs. 0.5%) [52]. Unfortunately, there are no other larger randomised trials to support this nonsignificant difference and the experience in contemporary registries suggest similar lower percentages for both devices.

Although stroke risk appears highest in the first 24 h after TAVI, CVEs are also reported in the months following the procedure [53]. Bioprosthetic valve thrombosis has been considered a possible cause of delayed stroke and full anticoagulation in the first period after TAVI has been considered. In the GALILEO trial [54], thromboembolic events were higher in patients treated with low-dose aspirin and low-dose rivaroxaban, despite subclinical leaflet thrombosis was lower in this group. Therefore, delayed stroke aetiology after TAVI remains poorly characterized and is probably multifactorial [55].

Cerebral embolic protection devices are designed to avoid embolization to cerebral arteries during TAVI. These devices protect the ostium of the supra-aortic branches in the aortic arch. Their main features are procedural stability, filter capability and the ability to preserve the integrity of the aortic arch wall.

These filters are positioned across the origin of supra-aortic vessels before the advancement of the TAVI delivery system across the aortic valve and are retrieved at the end of the procedure. Device deployment could be challenging, when atherosclerotic plaques are closely located to the ostium of supra-aortic vessels, with the risk of plaque disruption and consequent cerebral embolization.

CPDs can be classified in capturing filters or deflectors: capturing filter devices can retain embolic material while deflector devices reject it towards the descending aorta [44].

### 3.1. SENTINEL™ Cerebral Protection System

The Sentinel embolic protection device (Boston Scientific, Marlborough, MA, USA) is designed to capture atherothrombotic debris during TAVI and is the first FDA-approved cerebral protection device. The system consists of two cone-shaped polyurethane filters (pore size 140 µm) mounted on a 6 French sheath. The device is deployed through the right radial access to the brachiocephalic and left common carotid arteries. The proximal filter (length 4.0 cm) is delivered into the brachiocephalic artery, covering all areas of the brain supplied by the right vertebral and right carotid artery and the distal filter (length 4.5 cm) is deployed in the left common carotid artery. The left vertebral artery is not covered by the Sentinel device and a small study [56] showed that a second filter placed in the left vertebral artery contained debris in an equal amount of patients as the Sentinel filters. So, if the Sentinel device is used, a second filter (e.g., the Wirion Embolic Protection device [Cardiovascular Systems Inc., St. Paul, MN, USA]) could be placed in the left vertebral artery for complete cerebral coverage.

The main limitation of the Sentinel system is its lack of different available sizes, matching the different aortic anatomies. The diameter of the supra-aortic vessels must be previously measured by a CT scan. Target vessel diameters are 9–15 mm for the proximal filter and 6.5–10 mm for the distal filter with an articulating sheath length between the two filters of 4.0 cm.

Sentinel is the most widely used CPD and is usually deployed a few minutes before starting the TAVI procedure and is withdrawn soon after into its own catheter [44,55].

The first generation of Sentinel was evaluated in two randomised controlled trials (MISTRAL-C and CLEAN-TAVI). In the MISTRAL-C [57], 65 patients with high surgical risk were randomised 1:1 to transfemoral TAVI with or without the use of the device. Borderline significant fewer new lesions with a smaller total volume, as assessed by magnetic resonance imaging (MRI), and a significant higher neurocognitive deterioration was found in the control arm. The CLEAN-TAVI trial [58] confirmed, in 100 patients, a smaller number of new lesions and lower volume lesions in the filter group. The incidence of any complication was similar in the two groups.

The multicentre, prospective, randomised SENTINEL trial proved both the device safety and the capability of capturing embolic debris in 99% of the 363 patients. However, there was no change in neurocognitive function and the reduction in new lesion volume on MRI was not statistically significant [59].

Although there is no doubt about the safety of the Sentinel system, the above-mentioned trials were not adequately powered for clinically relevant outcomes. PROTECTED-TAVI (NCT04149535) is now enrolling more than 3000 patients randomised 1:1 to TAVI with or without Sentinel system to evaluate the immediate post-procedural rate of stroke at 72 h or discharge, and this will hopefully clarify this fundamental question.

### 3.2. Embol-X

Embol-X device (Edwards Lifesciences Corp., Irvine, CA, USA) is a filter initially designed for cerebral embolic protection during cardiac surgery. It is a polyester mesh (pore size 120 μm) mounted on a self-expandable nitinol frame that must be placed directly inside the ascending aorta. Its use has been tested in one small and prematurely terminated randomised trial, in which 30 high-risk surgical patients treated with transaortic TAVI was allocated in a 1:1 fashion to filter protection device or not. Although debris was found in all filters used, there was no correlation with the incidence of new cerebral lesions. Patients protected with the Embol-X had a significant reduction of lesion volume in the territory of the middle cerebral artery, the vertebral and basilar arteries, while the reduction of total cerebral lesion volumes was nonsignificant [60].

### 3.3. TriGuard™ Device

The TriGuard 3™ (Keystone Heart Ltd., Caesarea, Israel) is meant to deflect cerebral emboli during TAVI while allowing maximal blood flow to the brain. It provides full cerebral protection covering all the three branches of the aortic arch with a dome-shaped semi-permeable mesh on a nitinol frame that is placed through an 8-Fr femoral sheath and deflects particles larger than 140 µm (Figure 3).

The DEFLECT I [61] and DEFLECT II [62] prospective, multicentre, single-arm studies showed, in 36 and 14 patients respectively, the feasibility and safety of using the first and second generation of the device with a similar number of new cerebral lesions but decreased lesion volume compared to historical controls.

The multicentre, prospective, single-blind, randomised DEFLECT III controlled trial confirmed in 85 patients the safety of the second generation Triguard™ HDH and showed that the use of the device increased freedom from cerebral ischaemic lesions by more than 50% and reduced single and maximum lesion volume by about 40% [63].

The REFLECT trial (NCT02536196) will assess the safety and effectiveness of the TriGuard™ HDH and the last generation TriGuard 3™ CPD in a larger sample of transfemoral TAVI patients (approximately 500 patients are currently enrolled).

### 3.4. Wirion Embolic Protection System

Wirion is a nylon filter on a nitinol frame with 120 µm pore size designed for peripheral vascular intervention. Its use is specifically indicated during atherectomy in calcified lesions of the lower extremities. It requires a 6-Fr sheath and can be deployed on any 0.014″ guidewire, fitting vessels size from 3.5 to 6.0 mm.

The WISE multi-center, non-randomised, open-label, single-arm study enrolled 120 high-surgical risk patients who underwent carotid artery stenting using Wirion to prevent cerebral embolization. The primary end-point—a composite of major adverse cardiac and cerebrovascular events (MACCE) rate, including death, stroke, and myocardial infarction during the procedure and within 30 days—was significantly lower compared to historical controls (3.3% vs. 6.3%; *p* = 0.0008). Stroke rates were lower in the filter group than in the historical control group (2.5% vs. 4.6%; *p* = 0.18) and device success was achieved in 99.1% of cases [64].

### 3.5. Embrella Embolic Deflector Device

Embrella (Edwards Lifesciences Corp., Irvine, CA, USA), was one of the first dedicated devices for TAVI, designed for deflecting debris during valve implantation. It is an oval-shaped nitinol frame (length, 59 mm; width, 25.5 mm) covered with a porous polyurethane membrane (100-μm pore size) that is meant to be inserted through a right radial or brachial arterial access with a 6-French delivery system. The frame of the device has 2 opposing petals positioned along the greater curvature of the aorta, able to cover both brachiocephalic and left carotid artery ostia [65]. The PROTAVI-C Pilot Study [65] proved no clinical benefits and possible harms with the use of the device, by showing a higher burden of procedural cerebral microemboli–evaluated with transcranial Doppler-in the 42 patients protected with Embrella compared to the control group (12 patients), suggesting that the insertion of the device could be associated with microembolization. Incidence and number of new silent cerebral ischemic lesions were similar in the two groups, although the volume of cerebral lesions was smaller in the device arm.

The device is no longer available.

### 3.6. Novel Perspectives for Cerebral Protection during TAVI

Several new CPDs are currently under development or in the early analysis phase.

Point-Guard™ Dynamic Cerebral Embolic Protection (Transverse Medical Inc., Denver, CO, USA) is a deflection device which allows maximal coverage of all great arch vessels, safeguarding patient during TAVI or other left-side procedures. The device consists of a flexible nitinol frame with a dual-edge perimeter seal filter mesh that is designed to isolate completely the supra-aortic branch ostia and to adapt to different aortic arch anatomy. The Point-Guard is currently only for investigational use since no clinical data are available [66].

ProtEmbo^®^ Cerebral Protection System (Protembis, Aachen, Germany) is another deflection device meant to be deployed at the aortic arch roof to cover all three supra-aortic vessel branches. The device is inserted through a 6-Fr left trans-radial sheath and is constituted of a 60 μm pore sized mesh that is the smallest among all CPDs (available for clinical use and under study), protecting from smaller sized debris [66]. The PROTEMBO SF Trial (NCT 03325283) will assess the safety and feasibility of the device.

The Emblok Embolic Protection System with Modified Pigtail Catheter (Innovative Cardiovascular Solutions, Grand Rapids, MI, USA) is a capture filter designed to be placed in the ascending aorta and aortic arch via femoral arterial access (11-Fr sheath) and the delivery system contains a 4-Fr pigtail for aortogram. The device provides complete brain protection from embolization thanks to a conical filter made of polyurethane mesh with a pore size of 125 μm supported by a nitinol frame. Its use is possible only for ascending aorta length ≥ 9 cm and an ascending aorta or aortic arch diameter between 30 and 35 mm. A prospective, nonrandomised, multi-center, first-in-man study enrolled 20 patients and proved the safety and feasibility of its use. No MACCE occurred and post-procedural diffusion-weighted (DW)-MRI showed similar new lesion volume to other cerebral embolic protection devices. In a post-hoc analysis, a trend toward a significantly lower burden of new lesion volume was found when a fully protected procedure was achieved (full protection is defined when Emblok was open for pre-dilation, valve deployment and post-dilation) [67].

Captis™ Embolic Protection System (Filterlex Medical Ltd., Yokneam, Israel) and Emboliner™ Total Embolic Protection Catheter (Emboline Inc., Santa Cruz, CA, USA) are two novel generation filters designed to capture debris aiming for full embolic protection in the cerebral as well as in peripheral vessels. Preliminary results on 31 patients protected with Emboliner showed a 46% reduction of 30-day major adverse cardiac and cerebrovascular events–death, stroke and stage 3 acute kidney injury–compared with a 12% historical performance goal, the ability to successfully deploy and retrieve the device in all patients and the capability of capturing debris in 100% of filters analysed [68].

It is quite clear that all TAVI procedures generate debris that is destined to embolize to the brain and other organs. It remains to be proven that the efficacy of filters or deflectors is sufficient to improve outcomes. The trials underway appear sufficiently powered to confirm their usefulness.

## 4. Optimal Valve Positioning to Reduce Pace-Maker Implantation

Pacemaker implantation is certainly not a life-threatening complication and you may argue that a sudden deterioration of the atrioventricular (AV) conduction due to the damage induced by the expansion of the TAVI valve simply anticipates a natural evolution of an existing conduction disorder. With TAVI expanding to younger patients this argument appears somewhat hollow, however, and there is general agreement that we need to understand better mechanism, incidence, and predictors of conduction disturbances after TAVI.

### 4.1. Conduction Disturbances

#### 4.1.1. Mechanism

The aortic valve has close spatial proximity to the conduction system. The atrioventricular node (AVN) is in close proximity to the subaortic region with the His bundle running on the lower edge of the membranous septum in the left ventricular outflow tract (LVOT).

TAVI prostheses are inserted in an intra-annular position and, in contrast to surgical valves, require exerting pressure against the aortic annulus to maintain the stent frame in position. Slight oversizing is also necessary to secure the transcatheter heart valve (THV) and reduce paravalvular regurgitation; however, excessive oversizing can result in increased compression of the conduction system. During guidewire insertion, balloon pre-dilation, and valve deployment, mechanical damage to the surrounding tissue may develop and according to its cause (oedema, haematoma or necrosis of the conduction system components), the consequent disorders can be temporary or persistent. Almost half of these conduction abnormalities may improve over time and not require PPMI (permanent pacemaker implantation) due to resolution of the inflammation and oedema caused during the procedure [69] but also in these fortunate circumstances they affect recovery forcing the patient to a prolonged bed rest and in-hospital monitoring.

#### 4.1.2. Incidence

Conduction disturbances have been reported with varying and different incidences across studies, depending on the valve type, pacemaker implantation’s indication, which also varies between different centres and operators-and the population included. Indeed, the results of some studies may be skewed by the high rate of patients with pre-existing pacemakers. The frequency of pre-existing permanent pacemaker (PPM) also varies widely and appears to be correlated with patient age, comorbidities and surgical risk. In the PARTNER 1 and CoreValve US Pivotal trials, enrolling patients with high or prohibitive surgical risk, the prevalence of pre-existing PPM was 21% to 22% [70,71]. In contrast, in the PARTNER 3 and Evolut Low Risk trials, which enrolled younger patients with fewer comorbidities, the prevalence of PPM at baseline was between 2.0% and 3.4% [45,46].

According to the valve type, the incidence of both new-onset LBBB and PPMI are higher after implantation of the self-expanding CoreValve System (Medtronic Inc., Minneapolis, MN, USA) than of the balloon-expandable SAPIEN or SAPIEN XT systems (Edwards Lifesciences, Irvine, CA, USA) [69].

The PPM implantation rate of the Medtronic self-expanding CoreValve was initially reported to be between 25% and 35%, with a decreased incidence after the introduction of the new generation self-expanding CoreValve Evolut R, repositionable and therefore allowing a more precise high implantation. Self-expanding valves, however, have a PPM implantation rate substantially higher than SAVR (<5%) and TAVI using Edwards balloon-expandable SAPIEN valves (Edwards Lifesciences, Irvine, CA, USA) (5–10%) [52,72].

The higher rate of PPMI with self-expanding valves can be explained by the greater need for oversizing and the continuous radial force exerted by the self-expanding nitinol on the conduction system.

##### New-Onset Persistent Left Bundle Branch Block (NOP-LBBB)

The rate of new-onset LBBB after TAVI ranges from 4% to 65% with the rate of PPMI ranging from 2% to 51% [73].

Patients receiving a CoreValve demonstrate a substantially higher rate of new-onset LBBB (27%; range 9% to 65%) compared with those receiving an Edwards SAPIEN valve (11%; range 4% to 18%) [74].

New-onset LBBB occurs mainly during the procedure or within 24 h afterwards, though delayed presentation (after 24 h) is also possible. In the study by Testa et al., LBBB after TAVI occurred in 43.0% of 1060 patients treated with Medtronic CoreValve (MCV), but the incidence decreased to 27.3% at discharge and remained stable at 30 days [75].

Urena et al. reported the rate of new-onset LBBB to be approximately 20.0% after TAVI with Edwards SAPIEN Valves (ESV) and 50.0% of new-onset LBBB resolved within a few days after TAVI, leading to a rate of new-onset persistent LBBB of approximately 10% [76].

In another study, Franzoni et al. showed a higher incidence of LBBB following MCV (50.0%) than ESV (13.5%), which reduced at discharge to 32.2% for MCV and 8.6% for ESV, respectively [77].

Mainly half of the patients resolve their new-onset LBBB, but when persisting or evolving toward a high degree AVB a PPMI is requested.

In a recent meta-analysis, a higher rate of PPMI at 1-year follow up was observed among patients with new-onset LBBB, compared with those who did not develop LBBB. Overall, LBBB leads to an increased likelihood of new PPMI early after TAVI with a higher incidence of PPMI in the MCV compared with the ESV, as confirmed in a randomised controlled trial [78].

However, with increased operators’ experience and the reduction in implantation depth (ID) allowed by the improved delivery techniques, the frequency of LBBB after TAVI has decreased significantly over time, especially with MCV THVs.

##### Permanent Pacemaker Implantation

Complete AV block after TAVI is the most commonly reported indication for permanent pacing [79]. According to a recent systematic review, the overall rate of PPMI after TAVI with new generation valves ranged between 2.3% and 36.1% [80]. The early generation Medtronic CoreValve resulted in a higher risk of PPM implantation (range 16.3% to 37.7%), which remained relatively high with the newer Medtronic CoreValve/Evolut R valve (range 14.7% to 26.7%), whereas the Edwards SAPIEN 3 valve resulted in a lower risk (range 4% to 24%) [80]. More recently, several contemporary studies with the SAPIEN 3 and SAPIEN 3 Ultra valves have demonstrated new PPM rates as low as 4.4% to 6.5%, grossly comparable to the surgical risk [81]. Recent studies have also shown a reduction in the rate of new PPM with the self-expanding Evolut R and PRO valves (Medtronic, Minneapolis, MN, USA) to 10–20% or less [82,83].

#### 4.1.3. Predictors of Conduction Disturbances

The rate of conduction disturbances after TAVI is highly variable and is dependent on many pre-existing and intraprocedural factors. These risk factors can be categorized as clinical, electrocardiographic, anatomic, or procedural factors [84].

##### LBBB after TAVI

Clinical factors include the presence of preprocedural conduction abnormalities, especially prolonged duration of the QRS interval at baseline, female sex, previous coronary artery bypass graft, diabetes mellitus or aortic valve’s calcification as anatomic factors.

Procedural factors include CoreValve implantation, transapical access, pre-dilation, oversizing, and lower ID. The most consistently reported procedural characteristic associated with the occurrence of LBBB are valve type and ID within the LVOT [85].

While the external pericardial wrap in the newer generation self-expanding Evolut PRO (EP) (Medtronic Inc., Minneapolis, MN, USA) has helped to reduce paravalvular leaks (PVL), its interaction with LVOT anatomy has not been examined, and few studies have looked at predictors of NOP-LBBB in the newest self-expanding THV Evolut R (ER) and EP [86].

##### PPM Implantation after TAVI

A pre-existing right bundle branch block (RBBB), the use of a self-expanding THV, and ID below the aortic annulus plane have been included as the earliest recognized risk factors for new PPM after TAVI.

A meta-analysis by Siontis et al. [87] provides evidence for a number of variables that serve as predictors of PPM implantation after TAVI in high-risk patients. Male sex, pre-procedural evidence of abnormal AV conduction (including first-degree AV block, left anterior hemiblock, and RBBB), and intraprocedural AV block indicate an increased risk of PPM implantation after TAVI for patients receiving any type of prosthesis, although the risk of PPM implantation was 2.5-fold higher in patients receiving the MCV than in those receiving the ESV in an unadjusted analysis. These variables remained significant predictors of permanent pacing among patients with MCV bioprosthesis.

An analysis from the PARTNER trial identified the prosthesis diameter over-sizing, relative to the LVOT, and the left ventricle end-diastolic diameter, in addition to RBBB, as risk factors for PPM [88].

Multiple additional studies have now confirmed THV ID as the most important modifiable procedural predictor of PPMI with different prostheses.

Important new predictors of PPMI after TAVI have included THV over-sizing relative to the annulus or LVOT area, calcium location and burden, and membranous septum length (MSL). The latter serves as an anatomic landmark of the distance between the aortic annulus and the exit point of the bundle of His, where the conduction system crosses to the left side of the heart (Figure 4).

Hamdan et al. found that the difference between MSL and implantation depth is the single most powerful predictor of AV block and, together with calcification in the basal septum, the most powerful post-procedural predictor of PPMI with self-expandable valves [89].

Well-established predictors can be useful tools to guide clinical decision-making before and after TAVI to improve clinical outcomes. An appropriate device selection, the identification of patients at increased risk of PPM implantation after TAVI, and the decision for permanent pacing are mandatory to prevent AV-block-related complications, including syncope, exercise intolerance, heart failure, and sudden death.

#### 4.1.4. Implantation Techniques: Measures to Reduce the Risk of Pacemaker Implantation

Jilaihawi et al. [83] applied in a prospective cohort a meticulous valve implantation technique taking into consideration MSL (the so-called MIDAS [minimizing depth according to the septum] approach) and trying to avoid any mechanical interference with the His bundle. This determined a valve implantation strategy consisting of systematic measurement of MSL by pre-procedural CT scan and targeting a valve depth (as measured at the level of the noncoronary cusp) less than MSL.

Although previous studies have already highlighted the importance of MSL on conduction disturbances post-TAVI, the work of Jilaihawi et al. [83], through the implementation of a pre-determined valve positioning strategy according to MSL, reported one of the lowest rates of PPMI and new-onset LBBB post-TAVI. The total new PPMI rate was reduced from 9.7% to 3.0%, close to 6 times lower than the reported PPMI rate of the recently published Evolut low-risk trial [90].

A strategy that minimizes the risk of interaction of the conduction system with the bioprosthesis and a better understanding of the location of the conduction system relative to the aortic annulus basal plane was recently proposed. Using the cusp overlap technique, the noncoronary cusp (NCC), the most inferiorly oriented cusp in the LVOT, is isolated from the superimposed left coronary cusp (LCC) and right coronary cusp (RCC) (Figure 5).

Gada et al. [91] demonstrated that by isolating the NCC and overlapping the NCC/RCC commissure along the basal annular plane, the implantation view can be optimized during THV deployment. The cusp overlap view elongates the LVOT separating the conduction system from the annular plane and positions the NCC/RCC commissure in the centre of the fluoroscopic view by isolating the NCC. This view can be easily identified during preprocedural planning with CT reconstruction (e.g., 3Mensio, Pie Medical Imaging, The Netherlands). This fluoroscopic view may lead to a more precise implantation depth, thereby minimizing the risk of interaction with the conduction system.

### 4.2. Aortic Regurgitation Post TAVI

#### 4.2.1. Definition

Post-TAVI leaks can be divided into transvalvular, paravalvular, and “supraskirtal” leaks.

Transvalvular leaks are quite rare and are caused either by incorrect sizing of the valve or by a reduction in leaflet motion due to rupture or trauma during post-dilatation. Supraskirtal leaks are caused by a grossly incorrect low positioning of the valve that determines a blood passage above the valve plane, between the metallic meshes. They usually require an emergency correction with a valve-in-valve technique.

Finally, para-valvular leaks (PVL) are the most frequent and are characterized by a complex aetiology. In most cases, they are caused by one or more factors such as insufficient prosthesis’s adherence to the aortic annulus, valve sizing error, valve malposition, or suboptimal implantation.

The risk factors for post-TAVI PVL can thus be divided into (1) anatomic factors, associated with patient characteristics (patient-dependent factors), and (2) factors associated with the procedure itself (technical and/or operator-dependent factors).

#### 4.2.2. Mechanism

During TAVI, the native valve is not removed but crushed. Thus, a trivial regurgitation is not uncommon and has been reported in up to 25% of patients for both available types of percutaneous valves (self-expanding/balloon-expandable).

However, the definition of “clinically significant” valve regurgitation is not fully established yet. In the past, mild or moderate aortic insufficiency was considered a predictor of increased mortality, but this is questioned in more recent observations, while severe regurgitation could be life-threatening.

Evaluating the presence and severity of regurgitation should include an assessment of both central and paravalvular components, with a combined measurement of ‘total’ aortic regurgitation (AR) reflecting the total volume load imposed on the LV.

#### 4.2.3. Predictors of Post-Procedural Paravalvular Aortic Regurgitation

Multiple anatomical, procedural and post-procedural risk factors are identified for the development of PVL after TAVI [92].

##### Annulus Dimensions and Shape

Adequate sizing of the native annulus and LVOT is essential in preventing patient-prosthesis mismatch and PVL. Annular eccentricity itself does not predict PVL, but excessive annular calcium and its asymmetric distribution are important predictors of AR. Indeed, in cases of the extremely large annulus, for which available THVs need to be over-expanded, the eccentricity of the annulus is associated with PVL.

Thus, an accurate assessment of the aortic valve annulus before TAVI is mandatory to select the optimal valve.

##### Valvular and LVOT Calcium

Although some calcium on the native aortic valve is helpful to secure anchoring of the prosthesis to the annulus, the presence of excessive calcium and asymmetry of distribution can preclude appropriate positioning against the annular wall, thus resulting in PVL and the need for post-dilatation after valve deployment. Using CT assessment with 850 Hounsfield Unit (HU) threshold for detection, it is shown that PVL can be predicted with the quantification of LVOT calcification and with a calcium volume ≥ 235 mm^2^ [93].

##### Valve Type

Several studies showed that self-expanding first-generation valves were associated with a higher incidence of PVL. On the other hand, a multicentre English registry with 2584 patients showed an increased incidence of PVL in patients undergoing TAVI with balloon-expandable valves.

In the RESPOND study with the Lotus valve, a lower incidence of PVL was showed: no cases of severe regurgitation and only 0.3% of moderate regurgitation occur. The different technology of this valve is based on its own total repositionability and the presence of an “adaptive sealing” which allows complete adhesion of the valve also in unfavourable anatomical conditions. This valve has been recently withdrawn from the market due to complexities associated with the valve delivery system.

New valve technologies have been developed to reduce PVL incidences, such as the external pericardial wrap in the newer generation self-expanding Evolut PRO (EP), that allows greater adhesion of the valve to the aortic annulus, and the polyethylene terephthalate skirt in the SAPIEN 3 valve [94,95].

Recently, the SOLVE-TAVI trial [49], which randomised 442 patients undergoing TAVI, highlighted an equivalence between self-expanding and second-generation balloon-expandable valves at 30 days in the composite primary endpoint-including mortality, stroke, moderate-to-severe PVL, and pacemaker implantation. Even the incidence of individual endpoints, in particular moderate-to-severe PVL, was not significantly different between the two groups. In a meta-analysis of He et al. [96], which compared three new-generation self-expanding and balloon-expandable valves (SAPIEN 3, Evolut R, and Acurate neo), a significant reduction of PVL has been demonstrated with the SAPIEN 3 compared to the Acurate neo [97].

##### Sizing

An appropriate valve sizing is crucial to reduce the incidence of PVL, allowing a proper adhesion to the annulus and avoiding a rupture of the annulus and/or coronary occlusion. Nevertheless, a slight degree of oversizing could be useful to allow the prosthesis to be firmly anchored to the aortic annulus and promote its proper functioning to prevent prosthesis embolization [98].

The use of a parameter such as the cover index to evaluate the dimensions congruence of the prosthesis with respect to the annulus is debated.

#### 4.2.4. Implantation Techniques

##### Measures to Reduce Aortic Regurgitation

Many efforts have been directed towards reducing the prevalence of PVL, through better identification of patients at risk for developing PVL and improved prosthesis design. Over time the prevalence of PVL after TAVI has drastically decreased, as operators have acquired more experience. Several corrective measures have been proposed to overcome significant residual PVL following TAVI. The presence of severely calcified cusps of the native aortic valve might prevent complete apposition of the prosthesis with the annulus leading to a typical eccentric PVL jet. Here, balloon post-dilation is an option to reduce the degree of regurgitation by obtaining a better expansion of the prosthesis stent frame and better sealing of the paravalvular space if the THV has been deployed at the correct ID. Post-dilation is also the treatment option for patients with frame underexpansion as the reason for severe PVL with the use of self-expanding THVs [99].

Accurate positioning of the THV with respect to the native aortic annulus is critical for ensuring a successful procedure, whereas suboptimal deployment can result in incomplete apposition of valve and annulus or even worse in incomplete sealing by the pericardial skirt of the stent frame allowing a considerable diastolic backflow into the left ventricle. For malpositioned THVs with too shallow (‘too aortic’) or too deep (‘too ventricular’) implantation of the prosthesis, valve-in-valve implantation is a viable treatment strategy to reduce significant PVL and to prevent bailout cardiac surgery. The second valve can be deployed in a way that the sealing pericardial skirts of both valves overlap and that the second valve ensures sealing with the native valve annulus.

Finally, ID is a parameter to consider to prevent both PVL and conduction disturbances. Different approaches exist to calculate the ID [100] and a deeper ID leads to a significant increase in PVL and conduction disturbances, particularly with self-expanding valves.

The nitinol frame of the MCV might produce a higher pressure on the ventricular septum as compared with the stainless steel or cobalt-chromium frame of the ESV, leading to an increased risk of damaging the left bundle branch. Also, a higher rate of deeper implantation of the MCV (>5 mm from the aortic annulus) might partially explain the differences between the two devices.

For MCV, a prosthetic implant of 5–10 mm below the native aortic annulus should be checked by fluoroscopy [101].

Takagi et al. [102] demonstrated how a low implant (>3 meshes of stent below the virtual basal ring) is an independent risk factor for PVL (odds ratio 3.67), while Sherif et al. [103] found that an implant less than 9.5 mm from the non-coronary cusp is predictive of severe PVL.

In recent years, implantation techniques have therefore adapted to the results of the studies and valve deployment is currently recommended in a higher seat than in the past, to reduce both the appearance of post-procedural leaks and permanent PM implantations.

## 5. Valve-in-Valve and Valve-in-TAVI

Bioprosthetic valves (BV) are being extensively used for SAVR in the past decades. However, their limited durability over time along with the increasing general life expectancy results in a growing population of patients with degenerated surgical heart valve (SHV) bioprosthesis who are preferably managed with a less invasive transcatheter therapy: valve-in-valve transcatheter aortic valve implantation (VIV TAVI) (Figure 6).

### 5.1. Outcomes of VIV TAVI

The first large study that evaluated the outcomes of VIV procedures was the “Valve-in-Valve International Data Registry” (VIVID). Dvir et al. [104] collected data on 459 patients who underwent VIV TAVI with a balloon-expandable (Edwards SAPIEN, Edwards Lifesciences, Irvine, CA, USA) in 54% or a self-expanding device (CoreValve, Medtronic, Minneapolis, MN, USA) in 46% of the patients. The overall 1-year survival rate was 83.2%. The patients with BV stenosis had a worse prognosis than patients with BV regurgitation (30-day mortality rate was 10.5% vs. 4.3%, while the 1-year mortality rate was 23.4% vs. 8.8% with HR 3.07). This could be explained by an increased incidence of procedural complications such as left main obstruction or patient-prosthesis mismatch. The authors found small BV size, STS scores higher than 20%, and a baseline left ventricular ejection fraction of less than 45% as predictors of mortality in VIV patients. These results were confirmed by a metanalysis on 976 patients [105] and by the Swiss-TAVI Registry [106]. In the more recent 3 years follow-up of PARTNER II-Nested Registry 3/Valve-in-Valve study [107], TAVI for prosthetic aortic valve failure was associated with improved survival, valve haemodynamics and, importantly, sustained quality-of-life outcomes. In this prospective multicentre study, 365 patients, who underwent VIV procedures at 34 different sites, were enrolled. At 3-year follow-up no increased mortality was observed in patients who underwent VIV TAVI, there were only 5 repeat aortic valve replacements (1.9%) for aortic valve dysfunction at a median of 783 days post-VIV TAVI and patient symptoms gradually improved. This registry also showed a good valve performance, with no significant changes in transvalvular gradients, indexed effective orifice area (EOA), or total aortic regurgitation between 30 days and 3 years and improvement of left ventricular function.

### 5.2. Comparison of VIV versus Re-Do SAVR

Hirji et al. [108], in their large nationwide study, demonstrated that compared to re-SAVR patients, VIV-TAVI patients had significantly lower 30-day mortality (2.7% vs. 5.0%), 30-day morbidity (66.4% vs. 79%), and rates of major bleeding (35.8% vs. 50%). Valve-in-valve TAVI was also associated with a shorter length of stay and higher odds of routine home discharges (OR 2.11, 95% CI 1.61–2.78) compared to re-SAVR. Nalluri et al. [109], in their review, observed that VIV-TAVI was associated with lower rates of permanent pacemaker implantation, lower incidence of acute kidney injury (AKI)—probably due to the longer operative time and duration of cardiopulmonary bypass in patients undergoing surgery—higher vascular complications rates and less risk of major bleedings. However, data on kidney injury are controversial. Vrachatis et al. [110] found no significant differences in AKI events between VIV-TAVI and SAVR with a trend towards higher dialysis rates in the VIV-TAVI group. The incidence of stroke was not significatively different between VIV-TAVI and SAVR.

### 5.3. Procedural Considerations for VIV-TAVI

Optimal pre-procedural planning and procedural execution, through a step-by-step approach, are crucial for an optimal result of a VIV procedure. Successful VIV procedure is based on correct identification of the surgical valve and choice of an appropriate transcatheter valve implanted in a correct position [111]. The first critical planning step is to identify the surgical BV that needs to be treated with VIV. The BVs can be broadly divided into stented and stentless based on the presence or absence respectively of a plastic or metallic frame that supports the valve leaflets. Every surgical valve has its own design, dimensions as well as fluoroscopic appearance. The sewing ring of the bioprosthetic valve, which is sutured to the native aortic annulus, provides the most reliable rigid anchor to hold the TAVI valve in place, so it is important to know the relationship between the fluoroscopic markers and its location. Although stentless BVs lack rigid support for sewing rings, the original suture line between BV and native aortic annulus is still used as the anchor for THV. A valve-in-valve app tool (ViV Aortic by Dr Vinayak Bapat [112]), available at online app stores, has been developed to aid the interventionalist in choosing the transcatheter device suitable for various surgical devices. It provides true internal diameters for every device (that can differ from the labelled diameter), its fluoroscopic appearance and proposes a target for implantation. For example, when the SAPIEN valve is used, the aim should be to place the lowest aspect of the THV 10–20% below the sewing ring of the bioprosthesis, while for the CoreValve, at least 4–6 mm below the sewing ring.

The stentless valves normally require a slight oversizing of the THV to achieve secure anchoring, since they lack both heavy calcifications of a native stenotic aortic valve or a rigid sewing ring of the stented devices. None of the commercially available stentless valves are radio-opaque, which makes the procedure challenging. Hence, techniques such as placement of multiple pigtail catheters at the base of the leaflets, multiple contrast injections and placement of a wire in the left main coronary artery are useful. Controlled deployment and the use of retrievable devices have facilitated this procedure.

### 5.4. Limitations and New Techniques

Despite advances in the diagnostic workup, prosthesis choice, and improvement in implanting techniques, VIV therapy has several potential pitfalls [113], namely: coronary obstruction, the risk of malpositioning, patient prosthesis mismatch and leaflet thickening/thrombosis.

### 5.5. Coronary Occlusion

The reported incidence is approximately 2% in recent studies [114]. It occurs due to the displacement of the leaflets of the surgical device towards the ostium of the coronary artery or towards the sinotubular junction, eventually causing hemodynamic instability. Several factors, such as anatomical features, i.e., coronary height, the distance between the device and the coronary ostia, dimensions of the sinotubular junction, or technical aspects (e.g., BV with leaflets mounted externally to the stent posts such as Trifecta-St. Jude Medical, St. Paul, MN, USA or Mitroflow, Sorin, Milano, IT valves), can explain this complication. New-generation, fully retrievable THV devices may be preferable if the risk of coronary occlusion is expected to be high. If the predicted risk of coronary occlusion is very high, coronary artery protection with a safety wire (normally also with a pre-loaded stent) is recommended. In selected cases with a near 100% probability of occlusion, a pre-stenting of the coronary ostia is performed (“chimney technique”) [115,116,117].

A novel interventional technique with splitting valve leaflets by means of an electrified guidewire (“Basilica technique”) has been successfully used both in native and prosthetic valves to reduce the risk of coronary obstruction [118].

### 5.6. THV Malpositioning

Bioprosthesis malpositioning occurs when THV is placed too low or too high, resulting in suboptimal hemodynamics or embolization. The main causes are suboptimal fluoroscopic landmarks in certain SHV that pose a challenge in optimal THV positioning. With improved expertise of the operators and the introduction of repositionable and retrievable THVs its incidence has fallen from 8% to 4% [119].

### 5.7. Patient-Prosthesis Mismatch

The so-called ‘‘Russian doll’’ effect is the further reduction in the effective orifice area and persistent residual mean gradient ≥20 mm-Hg. This is especially true in an SHV with a manufacturer label size 21 mm or less or a stenotic SHV. Residual gradients have been shown to result in patient–prosthesis mismatch and affect survival.

The recently introduced novel technique that aims to reduce the incidence of patient–prosthesis mismatch is balloon valve fracturing. The first case of intentional fracturing of SHV was reported by Nielsen-Kudsk et al. [120] in 2015. It consists of a high-pressure expansion of a non-compliant balloon to fracture the sewing ring of an SHV to improve the expansion of a THV implanted in a small SHV. Balloon valve fracturing can be combined with BASILICA procedure in case of small anatomy with a high risk for coronary obstruction [121].

### 5.8. Leaflet Thickening and Thrombosis

Leaflet thickening and subclinical leaflet thrombosis (SLT) [122], whose incidence ranges from 0% to 40% [123], is related to the extent of THV oversizing relative to the implanted bioprosthesis, especially in small SHVs. This finding occurred more frequently in TAVI compared to SAVR at 30 days (13% vs. 5%, RR 2.64) but the differences were diminished at 1 year (28% vs. 20% RR 1.38) [124]. Several mechanisms may be responsible, including insufficient eddy currents to achieve optimal leaflet closing, increased blood stasis on THV leaflets, adjacent frame inside the surgical valve and under-expansion of the THV inside the small SHV. SLT is often seen as hypoattenuated leaflet thickening (HALT) on CT and can lead to reduced leaflet motion (RELM). HAM (hypoattenuation affecting motion) is defined by the presence of HALT and RELM simultaneously. The natural history of SLT is not well known. The timing of SLT after TAVI, resolution of thrombus without treatment, and short, medium, and long-term consequences of the SLT, as well as the best therapeutic strategies for prevention and treatment, are topics about which little is still known. Optimal sizing and anticoagulation, when necessary, could be potential strategies to prevent this complication.

### 5.9. Reintervention after TAVI: TAVI in TAVI

TAVI in TAVI has been performed since the beginning of the TAVI era mainly due to a malpositioning of a TAVI valve requiring second valve implantation to treat periprosthetic regurgitation. The first longer follow-up on a patient treated with TAVI in TAVI was reported in 2005 by Ruiz et al. [125]. A first larger cohort of patients was described by Witkowski et al. [126] in a review of 43 cases published between 2002 and 2013. They observed an overall success rate of TAVI-in-TAVI from 90% to 100%, resulting in favourable short and mid-term outcomes in patients with acute failure of TAVI without recourse to open-heart surgery (mortality rate of 0–14.3% at 30 days).

Recently reports [127] on patients in stable condition treated with re-do TAVI for THV degeneration are emerging. The latter clinical condition is likely to be more frequent in the future with even multiple valve implantations in the same patient. Coronary re-access after TAVI-in-TAVI may be challenging especially when THVs with the high frame, high skirt and supra-annular leaflets are being employed [128]. Novel TAVI implantation techniques aiming to alight the THV leaflets to native leaflets of the aortic valve (commissural alignment), similarly to surgical implantation technique, are emerging to minimize interference with coronary ostia of the first and possibly subsequent THVs [129].

## 6. Conclusions

TAVI has dramatically changed since its inception. Technique refinements were massive in the last years and will continue in the future. Procedures will become more and more tailored to patients’ individual needs, both in younger low-risk patients and in more complex cases, aiming to reduce all complications and improve outcomes.

## Figures and Tables

**Figure 1 medicina-57-00711-f001:**
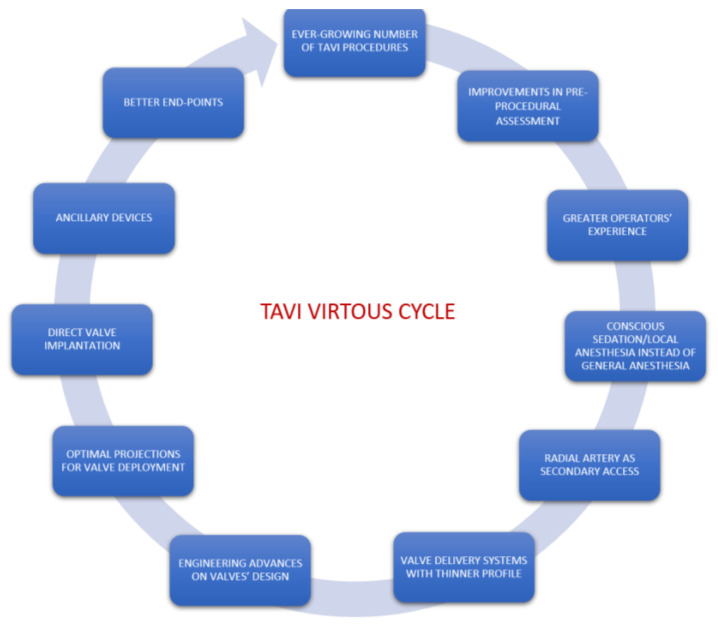
Transcatheter aortic valve implantation (TAVI) virtuous cycle generated by increased operators’ experience and improved technology.

**Figure 2 medicina-57-00711-f002:**
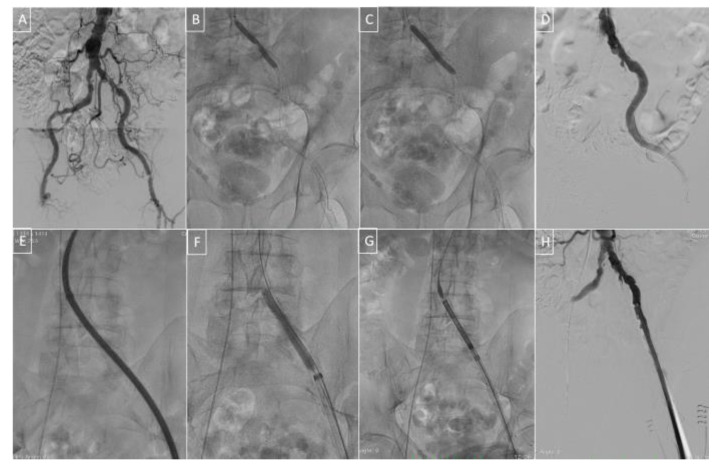
Intravascular lithotripsy (IVL) to facilitate TAVI (transcatheter aortic valve implantation) transfemoral access in a 79-years-old patient with severe aortic stenosis and end-stage chronic kidney disease on haemodialysis, arterial hypertension and diffuse peripheral arterial disease with claudication, previous renal artery stenting and severe carotid artery atherosclerosis. STS score: 7.5. Upper panels (**A**–**D**): severe occlusive calcific peripheral disease assessed by baseline aorto-iliac angiography (**A**); lesion preparation with IVL 6 × 60 mm Shockwave balloon on left common iliac artery inflated at 6 Atm with dog boning effect due to the severe calcification, 300 pulses delivered (**B**,**C**); intermediate result with greater lumen gain (**D**); lower panels (**E**–**H**): attempt to deliver a 14F sheath followed by further Shockwave 7 × 60 mm balloon inflation at 6 Atm (300 pulses) (**E**,**F**); the successful crossing of valve delivery system and final angiographic result showing absence of vessel rupture, dissection or perforation (**G**,**H**).

**Figure 3 medicina-57-00711-f003:**
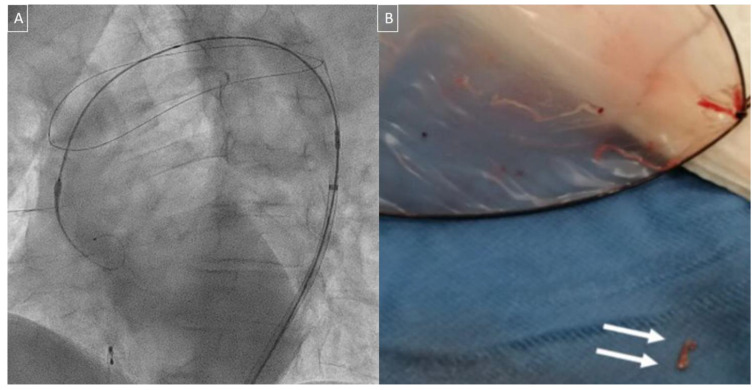
TriGuard™ cerebral protection device used in a patient with a bicuspid aortic valve during transcatheter aortic valve implantation (TAVI) (**A**) and the debris captured (**B**).

**Figure 4 medicina-57-00711-f004:**
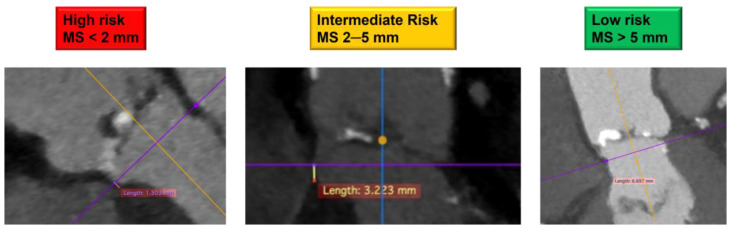
The impact of membranous septum (MS) length on permanent pacemaker implantation.

**Figure 5 medicina-57-00711-f005:**
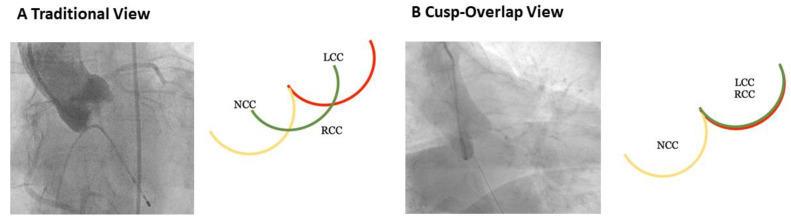
Optimizing fluoroscopic projections for transcatheter aortic valve implantation (TAVI). Panel (**A**) shows the basal aortography of the optimal projection for self-expandable valve implantation (“3 cusp view”), the nadirs of the three cusps (yellow = non-coronary [NCC], green = right-coronary [RCC], red = left-coronary [LCC]) are perfectly coplanar and equally spaced. Panel (**B**) shows an aortography of a cusp-overlap view obtained by overlapping the right-coronary cusp and the left-coronary cusp, isolating the non-coronary cusp and allowing to minimize interaction with the conduction system.

**Figure 6 medicina-57-00711-f006:**
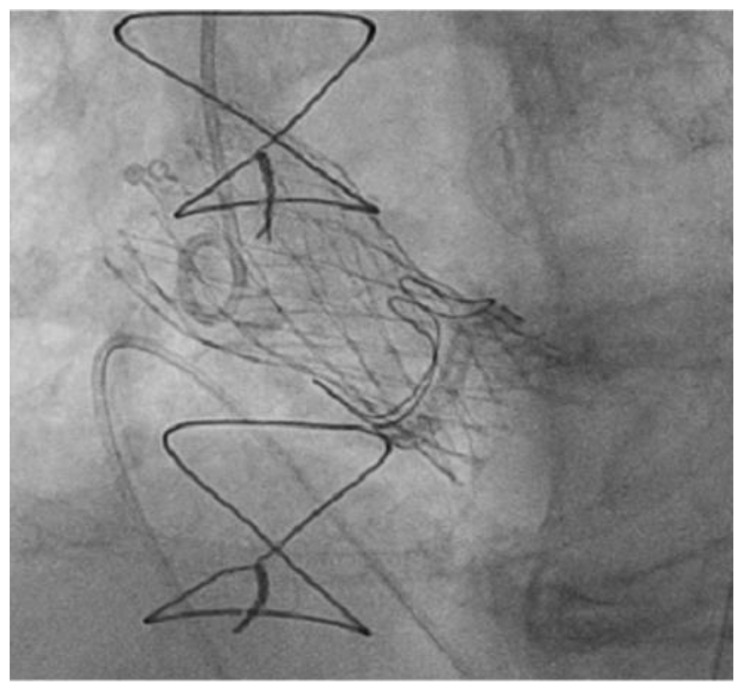
Post-implantation valve-in-valve view of a Corevalve Evolute R 26 mm inside a degenerated Sorin Mitroflow 23 mm.

**Table 1 medicina-57-00711-t001:** Cerebral Protection Devices. * Two vessel protection (brachiocephalic and left common carotid artery); ** Three vessel protection (brachiocephalic, left common carotid and left subclavian artery). Fr = French.

	SENTINEL™ Cerebral Protection System	TriGuard™	Embrella Embolic Deflector Device	Wirion Embolic Protection System	Embol-X	Point-Guard™ Dynamic Cerebral Embolic Protection	ProtEmbo^®^ Cerebral Protection System	Emblok Embolic Protection System with Modified Pigtail Catheter	Emboliner™ Total Embolic Protection Catheter	Captis™ Embolic Protection System
Mechanism	Capture	Deflection	Deflection	Capture	Capture	Deflection	Deflection	Capture	Capture	Capture
Access site and delivery approach	Radial artery–6 Fr	Femoral artery–9 Fr	Radial/brachial artery–6 Fr	Radial/brachial artery–6 Fr	Direct aortic–14 Fr	Unclear	Left radial–6 Fr	Femoral artery–11 Fr	Femoral artery	Femoral artery
Coverage	Partial protection *	Full protection **	Partial protection *	Partial protection *	Full protection **	Full protection **	Full protection **	Full protection **	Full protection (cerebral and peripheral vessels)	Full protection (cerebral and peripheral vessels)
Mesh pore size (µm)	140	140	100	120	120	-	60	125	-	-
Main evidence	MISTRAL-C; CLEAN-TAVI; SENTINEL	DEFLECT I-III	PROTAVI-C	WISE	Wendt D, Ann Thorac Surg 2015	-	-	Latib A, JACC Cardiovasc Interv. 2020	Pasupati S, J Am Coll Cardiol. 2020	-
Ongoing trial	PROTECTED- TAVR	REFLECT	-	-	-	-	PROTEMBO SF Trial	-	-	CAPTIS^®^ Study
Manufacturer	Boston Scientific, Marlborough, MA, USA	Keystone Heart Ltd., Caesarea, Israel	Edwards Lifesciences, Irvine, CA, United States	Cardiovascular Systems Inc., St. Paul, MN, USA	Edwards Lifesciences Corp., Irvine, CA, USA	Transverse Medical Inc., Denver, CO, USA	Protembis, Aachen, Germany	Innovative Cardiovascular Solutions, Grand Rapids, MI, USA	Emboline Inc., Santa Cruz, CA, USA	Filterlex Medical Ltd., Yokneam, Israel

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
