# Peer review of "Advancements in Transcatheter Aortic Valve Implantation: A Focused Update"

_medicina, 2021, doi:10.3390/medicina57070711_

Round 1
Reviewer 1 Report
Thank you for the opportunity to review this manuscript, which serves as a focused review on aspects of the transcatheter aortic valve implantation procedure.
With respect to the content covered by the review, it is comprehensive, concise and sound. There are a few comments I would like to make:
- It was not clear in the introduction as to the scope of the manuscript. There is not a discussion of the relative strengths of TAVR v SAVR as determined by randomised control trials. While this is acceptable in a manuscript discussing more technical aspects, it probably should be expressed early in the piece and it felt as thought the paper ended suddenly without this.
- Line 28 - I'n not ure it is right to say it is clear that it is "non-inferior even superior". Early follow up data is promising, however in Partner 3 the MACE curves appear to be converging after 2 years and longer follow up is critical to the understanding of outcomes in lower risk patients, who by definition are expected to live longer.
- Line 33 - Virtuous cycle -virtuous seems an overly dramatic term to use to describe the advances, and the diagram includes a few items that you wouldn't consider to be part of the cycle of advancement, eg anaesthesia, access site. But I appreciate the concept.
- 323 - this sentence is somewhat ambiguous - I assume it is meant that the device was associated with fewer lesions on MRI, and less neurocognitive deterioration?
- 436 - this paragraph is a little to informal and opinionated. While the ideas are appropriate to be discussed, perhaps it should use more subtle language.
- 665 - Why was the lotus device withdrawn? Was it indeed unfortunate, or were there other factors at play here?
- 842 - leaflet thrombosis (detected on CT as 'HALT') has been topical recently and probably deserves more discussion. It appears to be a throwaway line, and the scant attention provided may leave a reader implying a bias of the pro-TAVI authors.
Other than those brief comments, the paper is overall sound and worth reading for a comprehensive update in the technical aspects of the procedure.
Author Response
Thank you for the opportunity to review this manuscript, which serves as a focused review on aspects of the transcatheter aortic valve implantation procedure.
With respect to the content covered by the review, it is comprehensive, concise and sound. There are a few comments I would like to make:
Comment 1: It was not clear in the introduction as to the scope of the manuscript. There is not a discussion of the relative strengths of TAVR v SAVR as determined by randomised control trials. While this is acceptable in a manuscript discussing more technical aspects, it probably should be expressed early in the piece and it felt as thought the paper ended suddenly without this.
Reply to comment 1: Thank you for your revision and advice. As suggested, we added a paragraph at the end of the introduction to clarify the scope of the manuscript.
Comment 2: Line 28 - I'n not ure it is right to say it is clear that it is "non-inferior even superior". Early follow up data is promising, however in Partner 3 the MACE curves appear to be converging after 2 years and longer follow up is critical to the understanding of outcomes in lower risk patients, who by definition are expected to live longer.
Reply to comment 2: The primary endpoint of the PARTNER 3 Trial – death, stroke or rehospitalization - was significantly reduced after TAVI compared to SAVR also at 2 years of follow-up, even if differences in death and stroke were not statistically significant at 2 years. We modified the sentence as recommended.
Comment 3: Line 33 - Virtuous cycle -virtuous seems an overly dramatic term to use to describe the advances, and the diagram includes a few items that you wouldn't consider to be part of the cycle of advancement, eg anaesthesia, access site. But I appreciate the concept.
Reply to comment 3: At the beginning of TAVI procedures, general anaesthesia was the gold standard. The use of conscious sedation and local anaesthesia instead of general anaesthesia reduced procedural time and the need for inotropic and vasopressor. Similarly, the use of radial access as a secondary access instead of an additional femoral access reduced the risk of complications, so both items represent improvements of TAVI technique. We updated the diagram so that it is more clear that the items are part of the advancement of the technique.
Comment 4: 323 - this sentence is somewhat ambiguous - I assume it is meant that the device was associated with fewer lesions on MRI, and less neurocognitive deterioration?
Reply to comment 4: As recommended, we modified the sentence to clarify the concept: “The left vertebral artery is not covered by the Sentinel device and a small study showed that a second filter placed in the left vertebral artery contained debris in an equal amount of patients as the Sentinel filters. So, if the Sentinel device is used, a second filter (e.g. the Wirion Embolic Protection device) could be placed in the left vertebral artery for a complete cerebral coverage.”
Comment 5: 436 - this paragraph is a little to informal and opinionated. While the ideas are appropriate to be discussed, perhaps it should use more subtle language.
Reply to comment 5: As requested, we modified the paragraph removing subjective comments and inserting objective data.
Comment 6: 665 - Why was the lotus device withdrawn? Was it indeed unfortunate, or were there other factors at play here?
Reply to comment 6: The last January 2021 Boston Scientific Corporation (NYSE: BSX) has announced it has initiated a global, voluntary recall of all unused inventory of the LOTUS Edge™ Aortic Valve System due to complexities associated with the product delivery system. The voluntary recall is related solely to the delivery system, as the valve continues to achieve positive and clinically effective performance post-implant with no safety issue for patients who currently have an implanted LOTUS Edge valve. Given the additional time and investment required to develop and reintroduce an enhanced delivery system, the company has chosen to retire the entire LOTUS product platform. The complexity of the delivery system, manufacturing challenges, the continued need for further technical enhancements, and current market adoption rates led Boston Scientific Corporation to the difficult decision to stop investing in the Lotus Edge platform.
Comment 7: 842 - leaflet thrombosis (detected on CT as 'HALT') has been topical recently and probably deserves more discussion. It appears to be a throwaway line, and the scant attention provided may leave a reader implying a bias of the pro-TAVI authors.
Other than those brief comments, the paper is overall sound and worth reading for a comprehensive update in the technical aspects of the procedure.
Reply to comment 7: We modified the paragraph by adding a more detailed discussion on the topic.
Reviewer 2 Report
The Review "Advancements in Transcatheter Aortic Valve Implantation: a Focused Update" by Ciardetti et al. describes the current TAVR experience in a most complete state of the art manner.
There are only minor issues:
- Line 30: I feel that the statement "with no need for rehabilitation [1,2]" might give the wrong message. While TAVR is becoming safer, thus requiring less monitoring, rehabilitation addresses more than just safety. Admittedly, the role of rehabilitation in TAVR is not as well elucidated as in other areas of heart disease, but the presumption of its futility might be misplaced (e.g. compare: Cardiac rehabilitation after myocardial infarction in the community; Brandi J. Witt , Steven J. Jacobsen , Susan A. Weston , Jill M. Killian , Ryan A. Meverden , Thomas G. Allison , Guy S. Reeder , and V.éronique L. Roger; J Am Coll Cardiol. 2004 Sep, 44 (5) 988–996;
The role of cardiac rehabilitation in patients with heart disease; Sean R.McMahonaPhilip A.AdesaPaul D.Thompson; https://doi.org/10.1016/j.tcm.2017.02.005)
- Line 99: While John Webb is the senior author of [17], the mention of Toggweiler S. seems more appropriate as first, he is the first author and second, the citation does not mention John Webb by name, i.e. ... et. al.
- Line 165: "At present, US guided puncture approach is underused and adopted only in a minority of cases in clinical practice."; please provide appropriate citation
- Line 174: "This is not a rare condition, if considering that peripheral stenoses, tortuosity and vessel calcifications affect 35% of the elderly population 175undergoing TAVI procedures"; please provide appropriate citation
Just as a thought:
I understand that this is a review on TAVR. However, it might be worth mentioning the problematic implications for SAVR. As mentioned in the introduction, as the TAVR virtuous cycle progresses, this is at the cost of SAVR (, thus inducing a vicious cycle?): with less and less experience, who will be able to treat us properly, when we are "the selected cases" (compare line 88 ff. "...even if surgery will remain the gold standard in selected cases (mechanical valves, endocarditis, combined surgery on other valves, aortic aneurysms, severe coronary artery disease and unfavourable valve/access anatomy).
Author Response
The Review "Advancements in Transcatheter Aortic Valve Implantation: a Focused Update" by Ciardetti et al. describes the current TAVR experience in a most complete state of the art manner.
There are only minor issues:
- Line 30: I feel that the statement "with no need for rehabilitation [1,2]" might give the wrong message. While TAVR is becoming safer, thus requiring less monitoring, rehabilitation addresses more than just safety. Admittedly, the role of rehabilitation in TAVR is not as well elucidated as in other areas of heart disease, but the presumption of its futility might be misplaced (e.g. compare: Cardiac rehabilitation after myocardial infarction in the community; Brandi J. Witt , Steven J. Jacobsen , Susan A. Weston , Jill M. Killian , Ryan A. Meverden , Thomas G. Allison , Guy S. Reeder , and V.éronique L. Roger; J Am Coll Cardiol. 2004 Sep, 44 (5) 988–996;
The role of cardiac rehabilitation in patients with heart disease; Sean R.McMahonaPhilip A.AdesaPaul D.Thompson; https://doi.org/10.1016/j.tcm.2017.02.005)
Thank you for your revision and comments.
Reply to comment 1: Since "with no need for rehabilitation" may be misleading and the concept we wanted to express is instead that TAVI compared to SAVR allows shorter recovery time, we modified the sentence as recommended.
- Line 99: While John Webb is the senior author of [17], the mention of Toggweiler S. seems more appropriate as first, he is the first author and second, the citation does not mention John Webb by name, i.e. ... et. al.
Reply to comment 2: We changed the author citation as suggested.
- Line 165: "At present, US guided puncture approach is underused and adopted only in a minority of cases in clinical practice."; please provide appropriate citation
- Line 174: "This is not a rare condition, if considering that peripheral stenoses, tortuosity and vessel calcifications affect 35% of the elderly population 175undergoing TAVI procedures"; please provide appropriate citation
Reply to comments 3 and 4: We provided a citation.
Just as a thought:
I understand that this is a review on TAVR. However, it might be worth mentioning the problematic implications for SAVR. As mentioned in the introduction, as the TAVR virtuous cycle progresses, this is at the cost of SAVR (, thus inducing a vicious cycle?): with less and less experience, who will be able to treat us properly, when we are "the selected cases" (compare line 88 ff. "...even if surgery will remain the gold standard in selected cases (mechanical valves, endocarditis, combined surgery on other valves, aortic aneurysms, severe coronary artery disease and unfavourable valve/access anatomy).
About your thought: I appreciate your interesting reasoning but personally I do not believe that the virtuous cycle that has characterized the evolution of TAVI in recent years is detrimental to SAVR technique. Actually, the latter is well defined unlike TAVI that had to grow and refine and the indications of SAVR, even if in selected cases, will still allow a maintenance of the good practice at least in larger centers.
Reviewer 3 Report
Authors need to highlight the facts about development of atrial fibrillation post TAVR as that can change anti platelet/ anti coagulation management especially for high risk population such as advanced kidney disease. They should consider discussing this fact in the background of recently published data.
Author Response
Authors need to highlight the facts about development of atrial fibrillation post TAVR as that can change anti platelet/ anti coagulation management especially for high risk population such as advanced kidney disease. They should consider discussing this fact in the background of recently published data.
Reply: Thank you for your revision and advice. The scope of our manuscript was to review the main technological advancements in TAVI procedure, as highlighted in the abstract. Antithrombotic management after TAVI is a topic beyond our scope that need a detailed discussion, especially in selected patients and in the background of recently published data. We included a brief mention of the topic in the section on cerebral protection devices for stroke prevention. The Galileo trial was already discussed in the same section.